# A Study on the Promotional Effect and Mechanism of National e-Commerce Demonstration City Construction on Green Innovation Capacity of Cities

**Jie Li \*, Shengjun Yuan and Jun Wu**

Business School, Guilin University of Electronic Technology, Guilin 541004, China
* Correspondence: lj19580768216@163.com

**Abstract:** Green innovation precisely reflects China's new development concept and is the general trend for urban development in the future. In addition, e-commerce might become an important breakthrough platform to promote urban green innovation. In order to explore the impact of the construction of national e-commerce demonstration cities on urban green innovation, this paper aimed to theoretically analyze the potential relationship between the two and the transmission mechanism. At the same time, we have examined the panel data from 297 prefecture level cities in China from 2005 to 2018 to explore the implementation effect of the pilot policies by using difference-in-differences and carried out a series of robustness tests. The results showed that: (i) The pilot policy of e-commerce demonstration cities exhibited a significant promotion effect on the green innovation capacity of the different cities, and in general, the promotion effect of the pilot policy is dynamically sustainable. (ii) Analysis of the further influence mechanism showed that the pilot policy could effectively promote the development of urban green innovation capacity by enhancing the level of urban informatization, thereby attracting the concentration of scientific and technological talents. This in turn can facilitate urban innovation and the entrepreneurship environment, among which the boosting effect of optimizing the urban innovation and entrepreneurship environment was the greatest. (iii) In terms of heterogeneity, the pilot policies showed significant positive effects on all the regions to which the cities belong, while the boosting effects were more significant for cities with large populations, non-central cities, and the general science and education cities. The findings of this study not only enrich the research results in the field of urban green innovation, but also have clear policy implications, which can provide useful guidance and reference value for the work of relevant departments.

**Keywords:** national e-commerce demonstration city; green innovation capability; multi-period double difference; triple difference

## 1. Introduction

At present, environmental destruction and resource depletion have evolved as serious problems that can affect the lives of people all over the world. In addition, as the old non-green development model has been criticized for hindering sustainable development. Different cities serve as important conduits of China's economic growth, industrial structure transformation, and progress and have a pivotal role in achieving regional high-quality development. In addition, as an emerging economy in transition, China has achieved rapid progress in areas related to industrialization and urbanization. However, at the same time, cities are caught in a dilemma of balancing economic growth and environmental protection, and a series of serious social concerns such as low efficiency, high energy consumption, and high pollution have emerged in recent years [1]. The state has attached great importance to the development of green innovation over the years. The outline of China's long-term goals for 2035 aims to accelerate green and low-carbon development and support green technology innovation (https://www.12371.cn/special/ssw2035/, accessed on 3 June

2022). The main essence of green innovation strategy organically integrates environmental protection and economic growth. It combines the dual benefits of technological innovation and environmental protection and can increase economic benefits while alleviating the pressure on resources and the environment [2], thus providing a new perspective for solving the urban development dilemma.

At present, with the continuous emergence of achievements in the field of information technology, the development of e-commerce has gradually appeared as a novel new driving force to promote China's economic growth. It can play a vital role in realizing the integrated development of new and traditional industries and enhancing the green and healthy development of the national economy. According to the data of the environmental impact report of e-commerce released in 2009, the energy consumption generated by online retail in China in 2009 was equivalent to 393 tons of standard coal, less than the sales volume of CNY 100 million yuan (traditional retail) (http://www.prcfe.com/web/cjb/2014-04/24 /content_1076117.htm, accessed on 5 June 2022). In recent years, e-commerce has developed more rapidly all over China. In 2018, China's total e-commerce transactions reached CNY 31.63 trillion, an increase of 7.3 times over 2009 (https://www.sohu.com/a/31832858 3_696946, accessed on 10 June 2022). Thus, in the critical period of China's economic transformation and because of the urgent need to adjust the industrial structure, harnessing the potential of e-commerce to promote regional economic growth and improving the level of technological innovation can serve as important means to promote regional economic transformation and development. This will not only help to significantly strengthen the level of regional innovation efficiency, but also can facilitate the transformation of the regional economic development mode from extensive growth towards green development.

In 2009, China began to pilot the first national e-commerce demonstration city in Shenzhen and gradually promoted it aggressively. This was achieved by emphasizing the integration of e-commerce into the various fields to enhance the vitality of economic development, improve the efficiency of resource allocation, curtail energy and material consumption, reduce environmental pollution, promote the innovative application of e-commerce in the different fields, and develop a green economy. In recent years, the integration of e-commerce and urban industries has played an important role, especially in the new energy industry and green supply chain [3], such as some new energy enterprises. For example, Xiaopeng Auto actively implements the operation mode of "new energy + e-commerce". In addition, in 2017, Jingdong Logistics launched the "Green Stream Plan" emphasizing green transformation in the whole process from production and packaging to recycling old products and promoting the innovative development of a green supply chain system. As of 2019, 230,000 tons of carbon emissions and 27,000 tons of disposable express waste have been reported to be reduced (http://news.haiwainet.cn/n/2019/1 230/c3541089-31690755.html, accessed on 17 June 2022). It can be seen that, on the one hand, the combination of e-commerce and related fields has played a positive role in the coordinated development of socioeconomic and environmental benefits. At the same time, the significance of the construction of national e-model cities fits the direction and goal of "innovation-driven + green development". The construction of these cities has become an important fulcrum to effectively promote the development of green innovation.

At present, the green economy is a hot spot for academic research, and green innovation is the key to significantly promote the development of the green economy. Green innovations can also be called eco-innovation and environmental innovation, which mainly implies the concept of sustainable development, taking into account both economic performance and environmental performance [4–6]. It can aid in improving resource use efficiency and reducing damage to the environment through green technology, process, organization, and system innovation, thereby achieving resource conservation as well as environmental friendliness [7]. Thus, green innovation practices have played a significant role in promoting China's carbon productivity [8] and high-quality economic development [9]. In relation to the influencing factors related to green innovation, most of the studies reported so far have been conducted at both the enterprise and city levels. For

instance, Deng et al. [10] found that energy-saving and low-carbon policies could markedly enhance the green innovation capacity of enterprises from the perspective of environmental regulation. Moreover, Wang et al. [11] reported that the introduction of the new Environmental Protection Law could significantly increase the pollution and illegal costs of enterprises, which in turn can reverse the green innovation of enterprises. In addition, some scholars have also explored the low-carbon city pilot perspective and found that the policy displayed a positive effect on the development of green innovation activities and green innovation development of enterprises [12]. We observed that the digital economy could enhance urban green innovation by significantly promoting economic agglomeration and optimizing the regional financial structure. Feng et al. [13] showed that both environmental regulation and foreign direct investment could exhibit significant promotion effects on urban green innovation. Li et al. [5] explored the potential impact of innovative city pilot policies on the level of green innovation in cities and found that the promotion effect brought by the policies implemented can last for about six years. However, some studies have also explored the negative factors related to the development of green innovation in cities. For example, Zhang et al. [14] found that environmental decentralization did not exhibit a favorable impact on the improvement of green productivity in the cities, and this impact was limited by the degree of freedom in the use of available government funds.

There are relatively few studies about the impact of e-commerce on green innovation and green economic development, and no unified conclusion has been reached so far. The facilitation theory argues that the use of Internet, the carrier underlying e-commerce, can effectively enhance the public's sense of efficacy in environmental governance and promote the public participation in environmental governance [15], and enterprises having e-commerce capabilities can promote innovative activities, improve productivity, and enhance their overall performance [16,17]. The disincentive theory argues that e-commerce can lead to problems such as excessive packaging and electricity waste [18], which can hinder the optimal development of a green economy. A number of studies have indicated e-commerce has both distinct advantages and disadvantages for environment and green development [19]. At the same time, these findings have primarily analyzed the possible relationship between e-commerce, the environment, and green development from a qualitative perspective and have not conducted quantitative research on the relationship between them. Second, there is also a lack of adequate research on the policy effects of national e-commerce demonstration cities, which is more reflected at the micro-enterprise level. There is also a paucity of urban and other macro level views about green innovation development. The construction of national e-commerce demonstration cities has injected substantial momentum into several economic indicators by attracting the inflow of foreign direct investment funds [20] and promoting green technological innovation in enterprises [21], advancing the rapid transformation of industrial structure [3], and enhancing green total factor productivity [22]. However, some scholars have pointed out that the output and application of green innovation activities should be viewed from a more macro perspective to obtain a clear picture [23]. Therefore, in the context of green economy development, the potential relationship between e-commerce and green innovation needs further empirical research.

To sum up, the research on green innovation in the existing literature primarily focuses on the concept, effects, and influence factors [5,6,9,10,14]. Moreover, existing studies have not paid enough attention to both national e-commerce demonstration cities and urban green innovation. Thus, due to the lack of integration between green innovation and e-commerce, existing studies are still unable to provide sufficient quantitative analysis and empirical evidence of the precise relationship between them. Therefore, this paper aimed to examine green innovation based on the city level and quantitatively evaluate the potential impact of the pilot cities on urban green innovation, which can aid to expand and deepen the research field of urban green innovation influencing factors. In addition, considering that Chinese cities differ substantially in terms of location, size, and administrative level, further heterogeneity analysis is conducted based on the difference characteristics found

among the cities, thus rendering the study of e-commerce model city pilot policies more detailed. Moreover, this study attempts to investigate the possible influence mechanism of e-commerce demonstration city pilots on urban green innovation capability from three distinct perspectives: city informatization level, technology talent concentration, and urban innovation as well as the entrepreneurship environment. Finally, from the perspective of practice, this paper puts forward a series of important policy suggestions and practical guidance for the construction of e-commerce demonstration cities to effectively promote urban green innovation.

The remainder of this paper is structured into different sections which are as follows: The second part describes the institutional background and research hypothesis of the article, whereas the third part is the setting of the benchmark regression model and the description of data and variables. The fourth part contains the analysis of the empirical results, the fifth part is the robustness test, the sixth part reports the mechanism analysis and heterogeneity test, the seventh part is the discussion of the article, and the eighth part encompasses the conclusion and overall policy implications of the article.

## 2. Institutional Background and Research Hypotheses

### 2.1. Institutional Background

In order to implement the role of e-commerce in the economy and society, the national development and Reform Commission, the Ministry of Commerce, and other departments have jointly launched the activity of "national e-commerce demonstration city". In 2009, the national development and Reform Commission and the Ministry of Commerce officially approved Shenzhen to create the first national e-commerce demonstration city pilot. In 2011, 22 different cities including Beijing, Shanghai, and Qingdao were approved as national e-commerce demonstration cities. Subsequently, according to the principle of "pilot first and gradual promotion", the national development and Reform Commission, the Ministry of Commerce, and other ministries and commissions approved several national e-commerce demonstration cities in various batches in 2014 and 2017, respectively. So far, a total of 70 cities in China have been approved as national e-commerce demonstration cities.

### 2.2. Mechanisms of Action and Research Hypothesis

The construction of a national e-commerce demonstration city is a vital policy to promote the development of e-commerce and to effectively promote the integration of e-commerce and cities at all the levels. It is also an exploratory practice for China to advocate economic and environmental integration and green innovation development. This paper hypothesizes that the construction of national e-commerce demonstration cities can not only directly promote urban green innovation, but also stimulate the development of urban green innovation through improving the level of urban informatization, attracting scientific and technological talents and optimizing the urban innovation and entrepreneurship environment. The specific mechanism and process involved are as follows:

2.2.1. E-commerce Demonstration Cities and Urban Green Innovation Capacity

The e-commerce demonstration city pilot can facilitate the effective integration of green innovation development elements. First, e-commerce is the integrated embodiment of a series of scientific and industrial technologies, and nowadays, all different kinds of industries and enterprises in the city are accelerating the layout of related aspects. The optimal development level of e-commerce is closely linked with enterprise technological innovation, and the two complement each other, while technological innovation can exert a positive effect on the green development of enterprises [24]. Second, the e-commerce development of the various industries and enterprises in the city can improve their intelligence level, increase their productivity, improve the efficiency of resource utilization, promote the transformation of the new and old kinetic energy of enterprises, and cultivate green industries, thereby driving the development of green innovation and generating ecological benefits [25]. Thereafter, the development of e-commerce technology and the

deep integration of the various industries and enterprises in the city can effectively improve the allocation of innovative resource elements of related industries and enterprises, which can aid the formation of green production and consumption patterns of enterprises. It can also facilitate the realization of green transformation, thus paving the way for rapid innovation to be able to promote the win–win situation of economic and environmental benefits of the city [26]. Therefore, to some extent, a national e-commerce demonstration city is primarily based on the development of e-commerce, with the help of major technological innovation in the field of information technology. In addition, through policy guidance as well as government support, it can attract innovative talents, capital, and other innovative factors to invest, reduce the transaction costs of all urban subjects, and improve the efficiency of resource allocation. Overall, it can cause marked improvement of regional economic development quality and lead to the emergence of various ecological benefits, thus promoting the level of green innovation in cities. In view of the above, Hypothesis 1 is proposed.

**Hypothesis 1.** *An E-commerce demonstration city pilot can promote the effective development of urban green innovation capacity.*

### 2.2.2. City Informatization and City Green Innovation Ability

E-commerce is one of the most advanced development forms of informatization, and under the policy guidance of national e-commerce demonstration city construction, the concept of e-commerce and e-commerce technology can penetrate into various fields of cities. This has been found to be conducive for promoting the improvement of the level of informatization cognition and informatization utilization of all urban subjects, advancing the process of urban informatization service and management. The improved level of urban informatization can effectively reduce the cost of inter-city communication and exchange, which can be conducive for enhancing urban innovation capacity [27]. Interestingly, Shi et al. [28] pointed out that the level of urban informatization could promote the efficiency of cities in informatization management and informatization services and thereby improve the scale effect of urban informatization activities, which can then benefit urban both industrial innovation and environmental governance. As a key carrier in e-commerce information technology, the Internet plays a pivotal role, and the level of development of the Internet can reflect the level of e-commerce development. This ranges from the initial simple Internet technology, Internet platform, and nowadays Internet ecology, which can significantly accelerate the innovation transformation of knowledge achievements, reduce the cost and risk of innovation activities, and improve the use and matching of innovation resources, all of which can have a positive effect on the regional innovation efficiency [29]. Han et al. [30] explored the possible influence of information technologies such as the Internet in driving regional innovation levels using the mediating effect model. The empirical results showed that information technologies such as the Internet could substantially strengthen the agglomeration effect of human capital and financial development in the process of information and knowledge dissemination and diffusion, which in turn can enhance the regional innovation levels. In view of the above, Hypothesis 2 is proposed.

**Hypothesis 2.** *E-commerce demonstration cities can positively influence the level of urban green innovation by improving the level of urban informatization.*

### 2.2.3. Clustering of Scientific and Technological Talents and Green Innovation Capacity of Cities

Scientific and technological talents constitute the most active factor of green innovation and the main source of green innovation development. The construction of e-commerce demonstration cities can be conducive to accelerate the building of innovative and creative talent platforms, and a large number of excellent information technology talents and creative talents have come in one after another, being enabled to provide human capital

support to enhance the city's green innovation activities. Innovative talents serve not only as the main body of knowledge output and technical innovation, but also an important carrier of knowledge and technology flow. Technological and creative talents can be transformed into the human capital of cities or enterprises, thus facilitating the technological innovation of enterprises and the contemporary development of the cities [31]. Specifically, human capital, on one hand, can have a positive impact on the autonomous innovation capacity of the cities, primarily in the sense that the concentration of human capital can produce a significant scale effect, which can facilitate the internal flow of knowledge, information, technology, and other innovation resources [32]. On the other hand, the concentration of human capital in the region can also be conducive to the mutual integration with the various industry partners and the reduction in investment of existing industries in intermediate resources. At the same time, high-quality human capital can also markedly improve the transformation efficiency of various industries in terms of the resource input, reduce the damage and pollution to the environment, and thereby effectively contribute to urban green innovation [6]. By constructing a model of creative talents' residential choice from the perspective of knowledge externality, Wang et al. [33] found that the concentration of the creative class people could significantly promote urban innovation and also facilitate the extension of the related industries to the middle and high end, thus closer to the resource-saving industries. In view of the above, Hypothesis 3 is proposed.

**Hypothesis 3.** *E-commerce demonstration cities can positively influence the level of urban green innovation by attracting the clustering of scientific and technological talents.*

### 2.2.4. Innovation and Entrepreneurship Environment and Green Innovation Capacity of Cities

The urban innovation and entrepreneurship environment is primarily reflected in the optimization of the urban innovation mechanism and the cultivation of an innovative atmosphere. Under the policy leadership of national e-commerce demonstration cities, on one hand, e-commerce has accelerated the improvement of the rights and protection system of all subjects in the city, among which talents act as the main body of innovation and entrepreneurship. However, it is crucial to break the institutional environment that can hinder innovation and entrepreneurship, including timely abolishing or altering the unscientific regulations on innovation, solidly promoting an institutional mechanism for the development of talent, improving the legal rights and protections related to intellectual property, etc. Li et al. [34] emphasized the positive role of the institutional environment on green innovation, and further, Chen et al. [35] specifically argued the potential ability of formal and informal systems to promote the collaborative innovation between different enterprises, local institutions, and agencies in industry–university–research. In addition, the improvement of laws and regulations such as the protection of useful knowledge products can bring into play the positive externality of intellectual property protection on green technological innovation, which is important for the proper allocation of social resources [36]. On the other hand, the development of e-commerce can play a vital role to deepen the reform in various fields such as finance. The development of e-commerce can effectively reduce information communication costs, approval costs, and transaction costs concerning the financial industry and reduce the financing risks of enterprises. It can also aid enterprises to cross the threshold of financing constraints, improve the innovation and entrepreneurship environment, release the innovation vitality of enterprises, and thus overall help to improve the green innovation capacity of cities [37]. In view of the above, Hypothesis 4 is proposed.

**Hypothesis 4.** *E-commerce demonstration cities can have a positive impact on the level of urban green innovation by optimizing the urban innovation and entrepreneurship environment.*

To sum up, the analysis framework of this paper is shown in Figure 1 below.

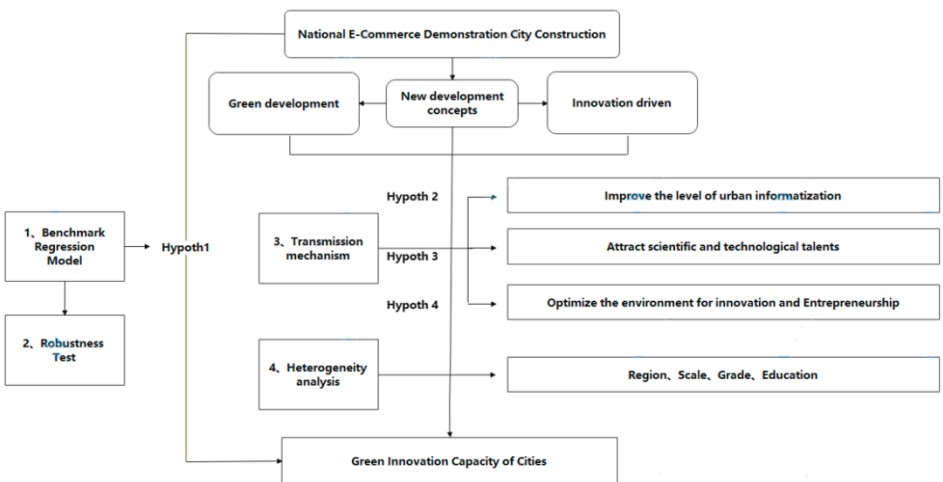

**Figure 1.** Analysis framework.

## 3. Description of Model, Data, and Variables

### 3.1. Model Setup

In this study, the national e-commerce demonstration city pilot is regarded as a quasi-natural experiment, and the sample of the prefecture-level cities is divided into two distinct groups (treatment group: belonging to the pilot city and control group: not belonging to the pilot city). Thereafter, a multi-period double difference model (DID) was used to identify the potential effect of national e-commerce demonstration cities on the green innovation capacity of cities with reference to the relevant research methods [10,12], as follows:

$$Gpatentper_{kt} = \alpha + \beta treated_k \times time_t + \gamma X_{kt} + u_k + \lambda_t + \varepsilon_{kt} \tag{1}$$

where $k$ and $t$ denote city and year, respectively. The explanatory variables serve as indicators reflecting the green innovation capability of the city. The $treated_k$ are city dummy variables indicating whether the city belongs to a national e-commerce demonstration city. They are set to one if the city belongs to, and set to zero otherwise. The $time_t$ are dummy variables for the year of e-commerce city implementation, set to zero for the year before belonging to the e-commerce demonstration city, and set to one if not. $X_{kt}$ are control variables including city economic development level (economy level), urbanization level (city level), foreign investment level (for invest), and education level (edu level); $u_k$ is city fixed, $\lambda_t$ is time fixed, and $\varepsilon_{kt}$ is a random error term.

### 3.2. Variable Description

The explanatory variable selected this paper was the city's green innovation capacity. Referring to the existing study [27], a natural logarithm of the number of green invention patent applications per 10,000 people in a city, and one was employed for the measurement. On one hand, it has been established that compared with green utility patents, green invention patents are more difficult to innovate and better reflect the level of substantive innovation. However, on the other hand, this study believes that green invention patents can serve as the most direct and convincing indicator for evaluating green innovation capacity compared with the use of parametric or non-parametric-based efficiency measures of green innovation. The explanatory variables in this paper are the cross term $treated_k \times time_t$ of the e-commerce demonstration city grouping dummy variable and the policy time dummy variable.

Moreover, another study by Luo et al. [6] showed that urban characteristic variables such as economic development level and foreign direct investment could also significantly affect urban green innovation in order to eliminate the influence of various other factors at the city level on the explained variables. Thus, the following variables were selected to control the different city characteristics: urbanization level (city level), foreign investment

level (for invest), education level (edu level), and economic development level (economy level). The urbanization level (city level) has been expressed as the ratio of urban built-up area to the total urban area. The foreign investment level (for invest) is expressed as the share of foreign direct investment in GDP. The education level (edu level) is expressed as the ratio of education expenditure to the total population of the region at the end of the year, and the economic development level (economy level) is expressed as the level of economic development expressed as the logarithm of per capita GDP.

The statistical description of the data of the main variables has been shown in Table 1. As seen in Table 1, the data of indicators reflecting the green innovation capacity of cities (gpatentper) had a minimum value of 0 and a maximum value of 2.460, and urbanization level (city level) had a minimum value of 0 and a maximum value of 97.18. In addition, foreign investment level (for invest) had a minimum value of $5.78 \times 10^{-7}$ and a maximum value. This indicated that the minimum and maximum values of both the data reflecting the green innovation capacity of the cities and other characteristic variables could vary greatly. Therefore, it is necessary to analyze the variability and the underlying mechanism of the influence of the construction of e-commerce model cities in the subsequent study.

**Table 1.** Descriptive statistics of the main variables.

| Variable | N | Mean | sd | Min | Max |
|---|---|---|---|---|---|
| gpatentper | 3177 | 0.438 | 0.505 | 0 | 2.460 |
| citylevel | 3177 | 9.164 | 9.693 | 0 | 97.18 |
| forinvest | 3177 | 0.00841 | 0.0705 | $5.78 \times 10^{-7}$ | 2.040 |
| edu level | 3177 | 0.504 | 0.617 | $2.02 \times 10^{-5}$ | 18.09 |
| economy level | 3177 | 10.66 | 0.787 | 6.422 | 14.27 |

Note: The research sample used in this paper is based on the development of green innovation in Chinese cities from 2005 to 2018, and the data have been mainly obtained from national intellectual property rights and city statistical yearbooks. Due to the serious missing statistics of a few pilot cities, 227 cities in the control group and 70 cities in the experimental group were finally selected, and the missing values were linearly interpolated. The data were also subjected to a 1% tailing process to avoid extreme worth effects.

## 4. Analysis of Empirical Results

### 4.1. Baseline Regression

Based on the above analysis, this paper mainly used a two-way fixed-effects multi-period double difference model to identify the impact of e-commerce, thus demonstrating city pilot policies on urban green innovation capacity, and the benchmark regression results have been summarized in Table 2. As shown in Table 2, the double difference estimator β, which is the major focus of this study, was significantly positive at the 1% level, thereby indicating that the national e-commerce demonstration city pilot can significantly promote urban green innovation capacity. Thus, Hypothesis 1 proposed above was validated. Moreover, considering that city characteristic variables such as urbanization level and foreign investment level can also affect the change of urban green innovation capacity, this study included urbanization level (city level), foreign investment level (for invest), education level (edu level), and economic development level (economy level) in the order of (2) to (5) in Table 2 of these indicators. It was found that the estimated coefficients of the cross term $treated_k \times time_t$ were only slightly different, but all of them were significantly positive at the 1% level. Taking the regression results in column (5) of Table 1, the estimated coefficients of foreign investment level were positive but insignificant, while the rest of the control variables were significantly positive, and it was observed that the coefficients of the control variables were also consistent with the reality. These findings also suggest that the improvement of urbanization level, education level, and economic level can all have significant positive effects on the improvement of green innovation capacity of the cities.

**Table 2.** Baseline regression results.

| Variables | Gpatentper | | | | |
|---|---|---|---|---|---|
| | **(1)** | **(2)** | **(3)** | **(4)** | **(5)** |
| $treated_k \times time_t$ | 0.364 *** | 0.363 *** | 0.363 *** | 0.365 *** | 0.360 *** |
| | (0.05) | (0.05) | (0.05) | (0.05) | (0.05) |
| citylevel | | 0.005 *** | 0.005 *** | 0.005 *** | 0.003 ** |
| | | (0.00) | (0.00) | (0.00) | (0.00) |
| forinvest | | | 0.016 | 0.013 | 0.009 |
| | | | (0.02) | (0.02) | (0.03) |
| edulevel | | | | 0.079 ** | 0.075 * |
| | | | | (0.04) | (0.04) |
| economy level | | | | | 0.217 *** |
| | | | | | (0.03) |
| City Fixed | yes | yes | yes | yes | yes |
| Time Fixed | yes | yes | yes | yes | yes |
| N | 3177 | 3177 | 3177 | 3177 | 3177 |
| r2_a | 0.603 | 0.607 | 0.606 | 0.615 | 0.636 |

Note: (1) The values in parentheses have been clustered to city-level robust standard errors; (2) *, **, *** denote 10%, 5%, and 1% significant levels, respectively.

### 4.2. Parallel Trend Test

The double difference model used to assess the construction of national e-commerce demonstration cities presupposes that the parallel trend test was satisfied, i.e., the change trend of green innovation capacity of non-pilot cities and pilot cities should be parallel before the policy can be implemented. Referring to the existing study [11], as shown in Figure 2, the event study method was used to further test the model:

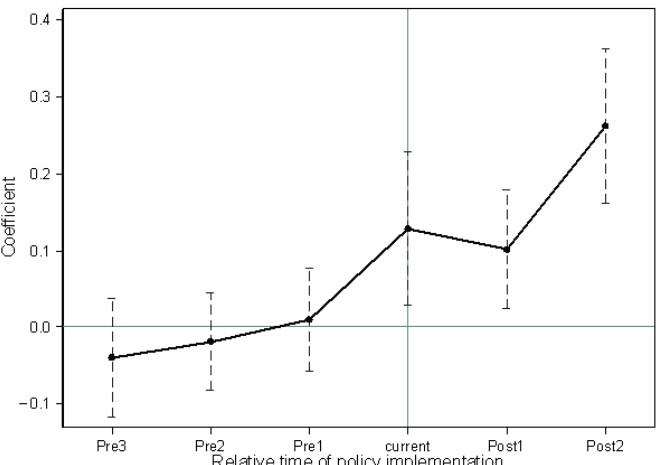

**Figure 2.** Parallel trend test results. Note: Regression coefficients in the figure have been expressed as relative values to the base period and clustered to the citylevel standard errors.

## 5. Robustness Test

The previous study suggested that pilot city policies can exert a significant effect on the green innovation capacity of cities; however, the robustness of this finding needs to be further tested. To overcome the interference of endogeneity issues such as omitted variables on the conclusion of the benchmark results, this paper aimed to analyze the multiple dimensions such as placebo test and exclusion of other policy effects to ensure the robustness of the conclusion (see Tables 4–6 for the results).

### 5.1. Placebo Test

In order to avoid the possible influence of unobservable value factors, as shown in Figure 3, the regression was continuously conducted by referring to the existing study [22]

by randomly simulating the sampling of the experimental and control groups in the target sample and at the same time, by avoiding the occurrence of small probability events by setting the number of sampling times. In this paper, 500 random samples were selected and the regression of the city brand index was performed according to the benchmark model. It can be clearly seen that the value of 0.360 estimated by the true model was significantly different from the distribution interval of the *p*-value, which indicated that the positive effect of national e-commerce demonstration city construction on the green innovation capacity of the city was not disturbed by other randomness factors, and the obtained results were robust in nature.

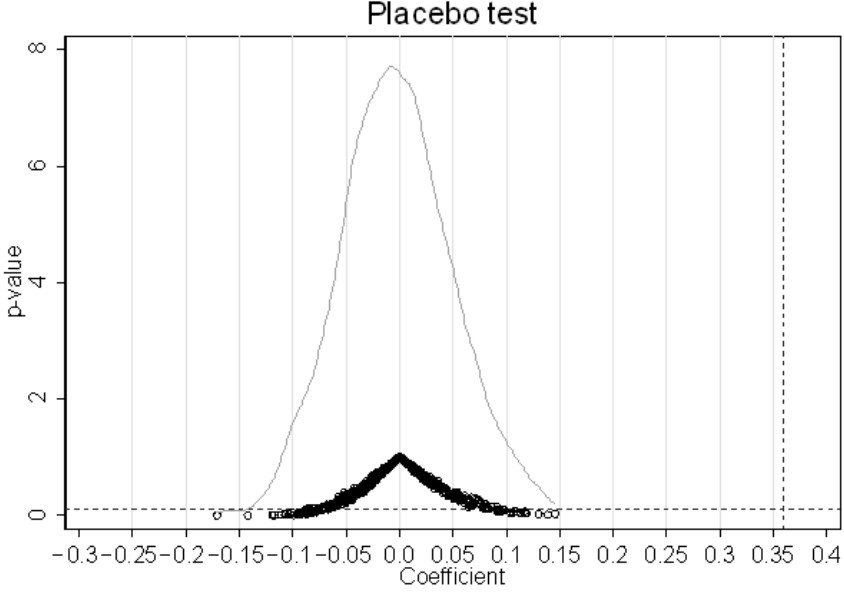

**Figure 3.** Placebo test.

*5.2. Instrumental Variables Method*

The sources of endogeneity problems can be diverse, and the instrumental variables approach can serve as a better strategy to effectively overcome endogeneity problems that are difficult to perceive such as two-way causality. However, the instrumental variables are often selected to satisfy certain prerequisites, namely, the relevance and exogeneity of the variables. Referring to the study by [22] and others, the number of telephone sets per 100 people at the city level in 1984 was selected as the instrumental variable for the core explanatory variables. First, the high number of telephones can to some extent reflect the better development of e-commerce in the region, thus supporting the correlation condition of this variable. On the other hand, the number of telephones per 100 people in 1984 might not directly affect the development of green innovation in cities due to its historical and invariant nature, thus supporting the exogeneity condition of this variable.

The regression results have been depicted in Table 3. As shown in Table 3 (second stage), the estimation results of the first stage indicated that the instrumental variables were significantly and positively correlated with the selection of national e-commerce demonstration pilot cities, thereby satisfying the requirement of correlation. The magnitudes of the Cragg–Donald Wald test statistics were significantly larger than the critical values, indicating that there was no weak instrumental variable problem. As depicted in the columns (1) and (2) of the Table 3, the regression coefficients of the implementation of e-commerce pilot policies on the development of green innovation capacity of cities with and without control variables were 0.464 and 0.459, respectively. This supported the core findings of this study, as shown in $treated_k \times time_t$, and both were found to be significantly positive at the 1% level.

**Table 3.** Instrumental variable method.

| Variables | Phase II | |
|---|---|---|
| | **(1)** | **(2)** |
| $treated_k \times time_t$ | 0.464 *** | 0.459 *** |
| | (0.052) | (0.054) |
| control | no | yes |
| City Fixed | yes | yes |
| Time Fixed | yes | yes |
| Cragg-Donald Wald F statistic | 7466.90 | 6583.52 |
| DWH | 14.448 | 11.6387 |
| | ($p$ = 0.0001) | ($p$ = 0.0007) |
| N | 4027 | 3180 |
| adj_R2 | 0.795 | 0.824 |

Note: (1) The values in parentheses have been clustered to city-level robust standard errors; (2) *** denote 1% significant level.

### 5.3. PSM-DID Test

Since sample selection bias can also contribute to the endogeneity problem of the model, this study employed the matching of the propensity scores of the experimental and control groups to reduce the interference caused by the sample selection problem on the regression results. As seen from Figure 4, the mean lines of the samples before and after matching were relatively closer; the gap between the experimental and control groups became significantly smaller, and the number of samples in the common region were increased, indicating that the matching was effective.

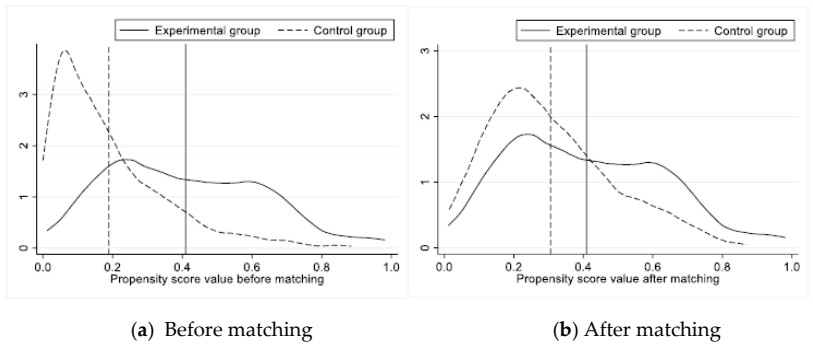

(**a**) Before matching                    (**b**) After matching

**Figure 4.** Kernel function plot before and after sample matching. (**a**) Before matching: (**b**) After matching.

In addition, a further balance test of the sample, as shown in Table 4, indicated that none of the covariates was significantly and systematically different after the matching.

**Table 4.** Balance test results.

| Variables | Processing | Mean | | Deviation | | *t*-test | | V(T)/V(C) |
|---|---|---|---|---|---|---|---|---|
| | | Experimental Group | Control Group | Deviation Rate | Deviation Reduction Ratio (%) | t | $p > t$ | |
| city level | Before matching | 12.75 | 8.0135 | 47.5 | 94.7 | 12.07 | 0.000 | 1.47 * |
| | After matching | 12.366 | 12.114 | 2.5 | | 0.44 | 0.662 | 0.76 * |
| for invest | Before matching | 0.00509 | 0.00969 | −7.9 | 87.7 | −1.56 | 0.119 | 0.01 * |
| | After matching | 0.00511 | 0.00454 | 1.0 | | 1.20 | 0.229 | 0.50 * |
| Edu level | Before matching | 0.47497 | 0.51395 | −5.8 | 69.4 | −1.52 | 0.128 | 1.78 * |
| | After matching | 0.44982 | 0.46175 | −1.8 | | −0.56 | 0.578 | 0.93 |
| economy level | Before matching | 11.161 | 10.5 | 89.7 | 98.6 | 21.73 | 0.000 | 1.02 |
| | After matching | 11.126 | 11.135 | −1.3 | | −0.27 | 0.787 | 1.09 |

Note: (1) *, denote 10% significant level.

With the above sample matching, the estimation of double difference was further carried out, and the specific results have been shown in Table 5. The difference between columns (1) and (2) was whether control variables were added or not, and it can be observed

that the estimated results were basically consistent with the baseline regression in terms of coefficient magnitude as well as direction, thus showing that the cross product term $treated_k \times time_t$ was significantly positive at the 1% level. Thus, the core conclusion of this paper was adequately supported.

**Table 5.** PSM-DID regression results.

| Variables | Gpatentper | |
|---|---|---|
| | **(1)** | **(2)** |
| $treated_k \times time_t$ | 0.320 *** | 0.320 *** |
| | (0.05) | (0.05) |
| control | no | yes |
| City Fixed | yes | yes |
| Time Fixed | yes | yes |
| N | 2842 | 2842 |
| r2_a | 0.792 | 0.807 |

Note: (1) The values in parentheses are clustered to city-level robust standard errors; (2) *** denote 1% significant level.

*5.4. Other Robustness Tests*

(1) Lags of explanatory variables: In the process of e-commerce demonstration city construction, there might be a significant delay in the policy effect or other situations where the effect may occur gradually over time. Therefore, this study lagged the core explanatory variables by one, two, and three periods, replacing the original variables in the model for regression, and the results have been shown in columns (1), (2), and (3) of Table 6. It was found that the cross term $treated_k \times time_t$ of the regression coefficients was significant and positive, which was the same as the results of the benchmark regression, indicating that the conclusions of the benchmark model in this paper were robust. (2) Changing the time interval of the sample: Since the third batch of e-commerce pilot cities occurred in 2017, which might alter the effect of policy evaluation, the sample interval of this paper was narrowed, and the regression was conducted within 2005–2016. The results are shown in column (4) of Table 6, and the regression coefficients of the cross term $treated_k \times time_t$ were found to be both significant and positive. (3) Next, in order to better study the net effect of policies on the construction of e-commerce model cities, this paper focused on the government-promoted innovative city pilot and low-carbon city pilot, which are the two major policies studied in this paper. In addition, this article specifically focused on these two policies, as these are more relevant, have a longer implementation period, and can exhibit crossover. Specifically, the above two types of policy dummy variables were introduced into the benchmark regression model, where dum_innov was whether the national innovative city pilot policy was implemented, and dum_lowcarbon was whether the national low-carbon city pilot policy was effectively implemented. The results are shown in columns (5), (6), and (7) of Table 6, and it can be observed that the regression coefficients of the core explanatory variables cross term $treated_k \times time_t$ were both significant and positive, and the results were still significant. (4) Avoiding the effects of extreme values: The 1% truncated tail operation was performed on the original sample, and the results have been shown in column (8) of Table 6. These findings provide support to the main conclusion of this paper.

**Table 6.** Robustness of the test results.

| Variables | (1) | (2) | (3) | (4) | (5) | (6) | (7) | (8) |
|---|---|---|---|---|---|---|---|---|
| $treated_k \times time_t$ | 0.336 *** | 0.297 *** | 0.228 *** | 0.377 *** | 0.291 *** | 0.328 *** | 0.267 *** | 0.221 *** |
| | (0.049) | (0.046) | (0.041) | (0.05) | (0.05) | (0.05) | (0.05) | (0.05) |
| dum_innov | | | | | 0.254 *** | | 0.246 *** | |
| | | | | | (0.05) | | (0.05) | |
| dum_lowcarbon | | | | | | 0.123 ** | 0.100 * | |
| | | | | | | (0.06) | (0.06) | |

**Table 6.** *Cont.*

| Variables | (1) | (2) | (3) | (4) | (5) | (6) | (7) | (8) |
|---|---|---|---|---|---|---|---|---|
| control | yes | yes | yes | yes | yes | yes | yes | yes |
| City Fixed | yes | yes | yes | yes | yes | yes | yes | yes |
| Time Fixed | yes | yes | yes | yes | yes | yes | yes | yes |
| N | 2927.000 | 2672.000 | 2420.000 | 2951.000 | 3180.000 | 3180.000 | 3180.000 | 3180.000 |
| r2_a | 0.817 | 0.829 | 0.843 | 0.802 | 0.651 | 0.639 | 0.654 | 0.480 |

Note: (1) The values in parentheses are clustered to city-level robust standard errors; (2) *, **, *** denote 10%, 5%, and 1% significant levels, respectively.

## 6. Mechanism Analysis and Heterogeneity Analysis

### 6.1. Mechanism Test

The theoretical analysis of this paper indicated that the construction of national e-commerce demonstration cities mainly has a profound effect on the development of urban green innovation through three distinct aspects. These include improving the level of urban informatization, attracting the concentration of the scientific and technological talents, and optimizing the urban innovation as well as the entrepreneurship environment, in order to validate the above-proposed mechanism. Referring to the test method [32], the following model was introduced.

$$Gpatentper_{it} = \theta_0 + \theta_1 treated_k \times time_t + \theta_2 treated_k \times time_t \times M + \theta_3 X_{it} + \varepsilon_{it} \quad (2)$$

Among them, *M* denotes the mechanism variables, mainly including the level of city informatization (net receive), the concentration of scientific and technological talents (scipeo), and the innovation and entrepreneurship environment (innov). The level of city informatization was measured based on the logarithm of the number of Internet broadband access subscribers, the concentration of the scientific and technological talents was measured by the ratio of the number of employees in the research and comprehensive technical service industry to the total regional population. In addition, both innovation and entrepreneurship environment were measured by the city innovation and entrepreneurship index (data from the China Innovation and Entrepreneurship Regional Index). The rest of the variables were consistent with the previous section.

The results in column (1) of Table 7 showed that the pilot cities could significantly improve urban green innovation capacity by improving urban informatization, as shown by the coefficient of $treated_k \times time_t \times netrecive$ that was significantly positive at the 5% level, and H2 Hypothesis was verified.

**Table 7.** Impact mechanism test.

| Variables | Gpatentper (2) | | |
|---|---|---|---|
| | (1) | (2) | (3) |
| $treated_k \times time_t$ | −0.457 | 1.143 *** | −2.939 *** |
| | (0.350) | (0.194) | (0.949) |
| $treated_k \times time_t \times$ netrecive | 0.157 ** | | |
| | (0.066) | | |
| $treated_k \times time_t \times$ scipeo | | 0.156 *** | |
| | | (0.041) | |
| $treated_k \times time_t \times$ innov | | | 0.744 *** |
| | | | (0.212) |
| control | yes | yes | yes |
| City Fixed | yes | yes | yes |

**Table 7.** *Cont.*

| Variables | Gpatentper (2) | | |
|:---:|:---:|:---:|:---:|
| | **(1)** | **(2)** | **(3)** |
| Time Fixed | yes | yes | yes |
| N | 3163 | 3159 | 3133 |
| adj_R2 | 0.811 | 0.814 | 0.805 |

Note: (1) The values in parentheses are clustered to city-level robust standard errors; (2) **, *** denote 5%, and 1% significant levels, respectively.

The results in column (2) of Table 7 indicated that the pilot cities can markedly improve their green innovation capacity by attracting a concentration of scientific and technological talents, as shown by the coefficient of $treated_k \times time_t \times scipeo$ that was significantly positive at the 10 level, and H3 Hypothesis was verified.

The results in column (3) of Table 7 suggested that the pilot cities can substantially improve their green innovation capacity by optimizing the urban innovation and entrepreneurship environment, as shown by the coefficient of $treated_k \times time_t \times innov$ that was significantly positive at the 10 level, and H4 Hypothesis was verified.

In summary, national e-commerce demonstration cities can partially mediate the improvement of urban green innovation capability by enhancing the level of urban informatization, attracting the pool of scientific and technological talents, and optimizing the innovation and entrepreneurship environment. The findings suggest that the construction of e-commerce demonstration cities can not only attract a number of scientific and technological talents to join, but also form the scale effect of the talents and promote the transformation of city informatization. At the same time, the construction of e-commerce demonstration cities can also improve the hardware and software as well as other infrastructure environments within the city, abolish the institutional system that hinders innovation and entrepreneurship, and thereby build an excellent urban innovation and entrepreneurship environment. Thus, it can effectively enhance the green innovation ability of the city.

*6.2. Heterogeneity Analysis*

A wide variety of heterogeneity exist among different Chinese cities in terms of the various resource endowments, education and culture, economic technology, and other factors. Therefore, it was next investigated whether differential degrees of influence of e-commerce model city construction on cities' green innovation capacity could be attributed to the aforementioned differences. Hence, this study considered the following factors to analyze the heterogeneity of the sample.

(1) City-region heterogeneity

This finding of this study indicated that compared to the non-eastern regions, eastern regions have certain advantages in terms of trade exchange, transportation infrastructure, and policy dividends. Considering the differences in the development of cities in the eastern, central, and western regions of China, it was analyzed whether the geographical location of these cities can contribute to the differences in the development of green innovation capacity of cities. Referring to the test method [6], the cities were first divided into two distinct zones, eastern and non-eastern. The regression results have been shown in Table 8, as shown in (1) and (2) of Table 8. It can be seen that the construction of the national e-commerce demonstration cities can play a significant role in promoting the development of green innovation capacity in both the zones, as shown by the positive and significant regression coefficient of the cross term $treated_k \times time_t$ at the 1% level.

**Table 8.** Heterogeneity analysis.

| Variables | (1) East | (2) Non-Eastern | (3) Type II | (4) Type I | (5) Central | (6) Non-Central | (7) Key | (8) Ordinary |
|---|---|---|---|---|---|---|---|---|
| $treated_k \times time_t$ | 0.325 *** | 0.374 *** | 0.175 | 0.324 *** | 0.114 | 0.321 *** | 0.095 | 0.285 *** |
| | (0.068) | (0.071) | (0.269) | (0.053) | (0.069) | (0.090) | (0.075) | (0.079) |
| control | yes | yes | yes | yes | yes | yes | yes | yes |
| City Fixed | yes | yes | yes | yes | yes | yes | yes | yes |
| Time Fixed | yes | yes | yes | yes | yes | yes | yes | yes |
| N | 1475 | 1705 | 1162 | 2018 | 389 | 2791 | 395 | 2785 |
| r2_a | 0.670 | 0.634 | 0.560 | 0.676 | 0.877 | 0.590 | 0.863 | 0.594 |

Note: (1) The values in parentheses are clustered to city-level robust standard errors; (2) *** denote 1% significant level.

(2) City size heterogeneity

As large-scale population could be conducive to the concentration of capital and talent, dissemination of information, knowledge, and other innovative resources. In view of the differences brought by these various factors, referring to the test method [20], cities with a population of three million are defined as Type I metropolitan cities and others are defined as Type II metropolitan cities, using the population of three million as the threshold. Columns (3) and (4) of Table 8 are reported, respectively, from which it can be observed that for the cities with large population size, the construction of e-commerce demonstration cities can significantly improve the green innovation capacity, as shown by the positive and significant regression coefficient of the model cross term $treated_k \times time_t$ at the 1% level. However, for the cities with small population size, the situation was found to be opposite. The possible reason could be that the city scale was too small to gather diversified innovation resources, and the spatial allocation efficiency of various resources was relatively low, which could lead to the insignificant effect of e-commerce demonstration city construction.

(3) Heterogeneity of city administrative level

In general, cities with high administrative level possess certain advantages in the policy dividends and resource allocation. Considering that there are high and low city hierarchies in China, it was next examined whether this can lead to differences in the development of green innovation capacity of cities. Referring to the test method [6], three types of cities, namely, municipalities directly under the central government, provincial capitals, and sub-provincial cities, are defined as the central cities and vice versa as non-central cities. The regression results have been shown in (5) and (6) of Table 8. It was found that the possible effect of e-commerce demonstration city construction on the central cities was not substantial. However, it exerted a significant promotion effect on the development of urban green innovation capacity in the non-central cities (0.321 > 0.114), primarily in the form of positive and significant at the 1% level for the regression coefficient of the model cross term $treated_k \times time_t$. The possible reason for this phenomenon could be that compared with the central cities, the non-central cities lack the advantages in policy support as well as resource allocation, and the urban green innovation capacity is relatively low. However, the potential of urban green innovation capacity enhancement is significantly greater in the non-central cities, which in turn can stimulate the vitality of urban green innovation factors, and thus the effect of policy influence on non-central cities is obvious.

(4) Heterogeneity of city science and education level

The high quality of science and education in the cities can cultivate superior-quality talent resources and reserve as well as provide talents for the development of cities. In addition, the number of colleges and universities in the cities and their popularity also play a pivotal role in the enhancement of cities' green innovation capacity. Based on the previous studies, cities with "211 project" universities are defined as the key science and education cities, and vice versa as ordinary science and education cities. The regression results have been reported in Table 8, columns (7) and (8) which show that the promotion effect of e-commerce model cities on key science and education cities was not significant, while the promotion effect on green innovation capability of general science and education

cities was at 1% significance level. The possible reason could be attributed to the fact that compared with the key science and education cities, the ordinary science and education cities do not possess the advantages of education resources but have a large potential to improve themselves, so the construction of e-commerce demonstration cities has a positive effect of "sending carbon in snow".

## 7. Discussion

In this study, we have evaluated the policy effect of the growing national e-commerce demonstration cities on urban green innovation. We noted that similar to the findings of Liu et al. [22], the construction of e-commerce demonstration cities in 2009 demonstrated that this policy can effectively promote green as well as high-quality development of cities, and the green and high-quality development was mainly concentrated in large-scale cities. From this perspective, the research conclusions are consistent in direction. However, since our research object is from the perspective of patents, it also emphasizes the final output, and to a certain extent, it included green high-quality development. Moreover, our study also indicated that the promotion effect is about 36%, which was significantly larger than the results of the above studies [22]. This difference can be attributed to the different measurement methods of the variables and data sampling. However, it may also lead to another problem, which is using patent data to measure completeness, which is also a drawback to be considered. Interestingly, this factor did not affect the starting point of this study but can intuitively reveal the promotion effect of e-commerce on urban green innovation to a greater extent.

In the relevant research on the evaluation of policy effects, heterogeneity analysis is the basic point that is paid the most attention. It has both the same points and obvious differences with the research of [5] Li et al. Although different policies can have a positive effect on urban green innovation in terms of the region to which the city belongs and the level of science and education, the promotion effect was different. The main reason may be that our research focuses on the positive effect of the integration of e-commerce at all the levels of the city from the perspective of informatization, which makes the results somewhat different. The research based on e-commerce also includes Gao et al. [3], Zhou et al. [20], and those reported by others. On the one hand, prior studies have paid more attention to the single dimension of the city, but this also brings enlightenment to our research because the level of the foreign investment and the transformation and upgrading of industrial structure in the city can also have a certain influence and impact on the green innovation of the city. However, the promotion effect of the foreign investment level in this study was not significant. On the other hand, this paper has creatively analyzed the transmission mechanism of e-commerce to urban green innovation from the perspective of urban informatization and has elegantly revealed the internal mechanism affecting urban green innovation.

Of course, it can also be noted that although our research indicated that the promotion effect was about 36%, as pointed out by the study of Bai et al. [37], urban green innovation might have a certain spatial spillover effect. Therefore, if the spatial spillover effect is not considered, whether the preciseness and accuracy of the results can be maintained remains a major question to be considered, which presents a new direction and challenge for future research. However, at present, our research findings are still scientific, novel, and reasonable.

## 8. Conclusions and Implications

### 8.1. Conclusions

This paper explored the potential impact of national e-commerce pilot city construction on the green innovation capacity of the cities using multi-period double difference and triple difference as a sample of green invention patent data of prefecture-level cities from 2005 to 2018. The findings indicate that, firstly, an e-commerce demonstration city pilot can significantly enhance the green innovation capacity of the cities. Second, from the

perspective of heterogeneity of factors such as city location, scale, administrative level, and science and education level, it can be concluded that an e-commerce city pilot can have a more significant impact on non-eastern cities and cities with large population size, but a lesser impact central cities and key science and education cities for city brand development. Third, from the perspective of influence mechanism, it was concluded that an e-commerce city pilot could promote the construction and development of urban green innovation capacity by substantially improving the level of urbanization information, attracting a large number of scientific and technological talents, and optimizing the urban innovation and entrepreneurship environment. The above findings have important policy implications for China and can aid in promoting the pilot e-commerce demonstration cities, firmly expanding the scope of e-commerce demonstration cities, and enhancing the green innovation capacity of cities.

Overall, in this study we have first summarized the experience of past e-commerce city pilots. The research results showed that status as an e-commerce city pilot could significantly improve the green innovation capacity of cities, and therefore the city pilot should be firmly promoted to realize the complete potential on urban green innovation. Moreover, in the process of expanding the scope of pilot cities, since the promotion effect of city pilots was more obvious in cities with large populations, more attention should be paid to the reasonable allocation of resources, and priority should be given to planning and supporting cities with large populations to maximize the policy dividends. As for the central cities and key scientific and educational cities, we should aim to strengthen the exchange and cooperation among cities, enhance cooperation between the cities, and promote the synergistic development of green innovation development among the cities in the region.

Second, we should strengthen the level of informationization of the main bodies of urban construction and consequently improve the management and service of urban informationization. The government can guide the various enterprises to increase the investment in enterprise informatization, improve the level of enterprise informatization, and actively promote the transformation of enterprise informatization. In addition, at the same time, the government can develop some preferential policies and measures to help enterprises during difficult situations. At the same time, the government also needs to actively carry out and hold various activities to educate the public about the various benefits brought by the improvement of individual informatization capabilities, update the non-informatization awareness of urban residents, and improve the convenience and satisfaction of the urban population. The government needs to pay proper attention to the digital transformation of the city and the work of informatization construction, focus on the interoperability as well as sharing of resources, and build more sound and efficient decision-making informatization management and coordination mechanisms. This can aid in integrating and advancing development by improving the comprehensive informatization capabilities of various enterprises, industries, city residents, and other stakeholders in the city, thereby effectively integrating all different types of information resources, optimizing the efficiency of information and other resource allocation.

Third, the city talent policy system can be improved and intake of high-tech talents in the e-commerce industry can be increased. Innovation is the first driving force, and talent is the first resource for rapid development. Attracting talents to gather, developing human capital in the cities, and mobilizing social talent resources can not only strengthen the main force of green innovation, but also promote the mobility of talents between the cities, reduce the cost of information exchange, and accelerate the knowledge dissemination among talents. It has been found that many cities have successively formulated and promulgated relevant innovative talent policies, implemented more active and open measures to attract talents, and at the same time built innovation platforms for scientific and technological talents, combined with the individual needs of research institutes and enterprises. They have also actively promoted targeted talent training programs, realized

the deep integration of industry–university–research, and delivered high-quality talent resources for local industries.

Fourth, urban infrastructure construction should be improved, and the soft environment of urban e-commerce innovation as well as the entrepreneurship mechanism should be better optimized. The cities should seize the opportunity to strengthen the construction of urban hardware and software and other infrastructure on the one hand, as high quality infrastructure can enhance the image of the city's attraction. On the other hand, government should actively act to improve the policy innovation and entrepreneurship guidance mechanism, reform incentives, focus on science and technology innovation systems, speed up the flow of talent, inject capital as well as other innovation resources, and innovate a talent innovation mechanism. It can also aid to optimize the investment environment, break the mechanism and system barriers of urban entrepreneurship and innovation, create a good business environment, stimulate urban innovation and entrepreneurship vitality, build excellent innovation and entrepreneurship ecology for entrepreneurs to implement green innovation in the city, and create good basic conditions and institutional environments.

*8.2. Limitations and Outlook*

The findings of our article clearly prove the promotion effect and influence mechanism of national e-commerce demonstration cities on urban green innovation from the aspects of theoretical elaboration, empirical analysis, and a robustness test. However, our study has certain limitations associated with it. First, the measurement method of urban green innovation used was relatively simple, and in the future, we can measure urban green innovation more accurately through index construction or other available methods. Second, a number of existing studies have suggested that urban green innovation might have spatial spillover effects, but this study does not consider the possibility of spatial spillover in the policy implementation, which may lead to evaluation bias. Therefore, future research can perform further analysis related to the measurement of urban green innovation or the spillover effect of urban green innovation.

**Author Contributions:** Conceptualization, J.L. and S.Y.; methodology, J.L. and J.W.; software, S.Y. and J.L.; validation, S.Y., J.L. and J.W.; formal analysis, S.Y. and J.L.; resources, S.Y. and J.L.; data curation, J.L. and J.W.; writing—original draft preparation, J.L. and S.Y.; writing—review and editing, S.Y.; visualization, J.L. and J.W.; supervision, S.Y.; project administration, J.L.; funding acquisition, S.Y. All authors have read and agreed to the published version of the manuscript.

**Funding:** NSFC "Research on the process and effect mechanism of regional branding—based on multi case qualitative research" (71862008); Humanities and social sciences fund of the Ministry of education, "Research on the process and effect mechanism of regional branding—a qualitative research method based on multiple cases" (18xja630007); Guangxi philosophy and social science project "Guangxi Regional branding application practice research" (17bgl004); Guangxi philosophy and Social Sciences project "research and application of brand value measurement based on Interbrand model" (ycsw2021168).

**Institutional Review Board Statement:** Not applicable.

**Informed Consent Statement:** Not applicable.

**Data Availability Statement:** Not applicable.

**Acknowledgments:** We thank Su Xin for helpful conversions.

**Conflicts of Interest:** The authors declare no conflict of interest.

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
