# Peer review of "A Study on the Promotional Effect and Mechanism of National e-Commerce Demonstration City Construction on Green Innovation Capacity of Cities"

_urbansci, doi:10.3390/urbansci6030055_

Round 1

Reviewer 1 Report

Manuscript ID: urbansci-1855165.

Title: A study on the Promotional Effect and Mechanism of National E-Commerce Demonstration City Construction on Green Innovation Capacity of Cities.

Authors: Lijie Li *, Shengjun Yuan, Jun Wu.

MAJOR COMMENTS:

The paper presents an important study about the relationship between national e-commerce demonstration city and green innovation capability in China. Moreover, it offers a sophisticated methodologies for empirical analysis (regressions and so on).

However, the text has a set of problem that is presented in detail in Minor Comments (bellow). This set compromised the overall quality of the text. Therefore, the article needs an important review.

MINOR COMMENTS:

1. (Lines 42-49) Lack of references to support that piece of information.

2. (Lines 49-50) Absence of reference to support those data.

3. (Lines 59-62) The arguments presented there need better (more qualified) bibliographic references.

4. (Line 64) I cannot find the references presented there in the journal (Gao, Changchun, 2021; Jin, Huan, 2022).

5. (Lines 83-85) It needs better bibliographic references.

6. (Line 85) The reference “Huang, 2018” is not correct (I think it is “Huang, Li, 2018”).

7. (Line 89) Lack of the co-authors (Xue and Yang) in reference (“Wang, 2020).

8. (Line 117) “(Wang, Wei, 2022)” is not in References section.

9. (Line 119) “Benitez, 2018” is an incorrect reference. And “Fang, 2020” cannot be found.

10. (Line 128) There is not in References section (“Jinhuan, 2022”).

11. (Line 140) I didn't find the text (“(Yuan Runsong, 2016”) in the journal.

12. (Line 162) Wrong numbering of section. It should be “2.2”, not “1.2”. And the following sections have incorrect numbering.

13. (Lines 198-200) Aghion (1992) is an important reference, but it needs updated reference(s). Furthermore, that reference is incorrect – it has two authors, Philippe Aghion and Peter Howitt.

14. (Lines 253-254) The equation (1) has two references, but I cannot find these texts in respectively journals. Please, review.

15. (Lines 450-451) As such above, equation (2) has a reference (“Wang, Ruchi, 2021”) I cannot find.

Author Response

Dear Editor,

       We are glad that the article can be reviewed by the experts and received valuable opinions from the experts. We carefully studied the suggestions of the teachers and tried our best to improve the overall writing of the thesis. Next, based on the suggestions of the experts, we will state how we have revised and the current revision situation.Since the number of lines of the newly revised article is different from the old one, the following contents shall be subject to the number of lines of the revised article.

  1. The overall situation of the article revision:

       We comprehensively considered the opinions of various experts. On the one hand, we reconsidered the structure of the article, and on the other hand, we revised and supplemented the missing details.

  1. Details of article revisions:

       We have made some adjustments to the structure of the article and the content. All references in the article are numbered.

2.1. (Lines 42-49) Lack of references to support that piece of information.

       According to the content of the article, we added the corresponding references,which can navigate to line 39 of the newly revised article.

2.2. (Lines 49-50) Absence of reference to support those data.

       We added the source of data according to the content of the article,which are specifically located at line 42 of the newly revised article.

2.3. (Lines 59-62) The arguments presented there need better (more qualified) bibliographic references.

       Integrating the opinions of all experts, we adjusted the structure of the article and followed the logic of the article, and updated the references,which are specifically located at line 144 of the newly revised article.

2.4. (Line 64) I cannot find the references presented there in the journal (Gao, Changchun, 2021; Jin, Huan, 2022).

       This part is due to the previous translation problems, We have used numbers to mark this part. This part can be located to the 137 line of the new revised article.

2.5. (Lines 83-85) It needs better bibliographic references.

       Integrating the opinions of all experts, we adjusted the structure of the article and followed the logic of the article, and updated the references,which are specifically located at line 92 of the newly revised article.

2.6. (Line 85) The reference “Huang, 2018” is not correct (I think it is “Huang, Li, 2018”).

       This part is caused by the problem of previous translation. We have used serial numbers to mark this part. This part can be located to the 92 line of the new revised article.

2.7. (Line 89) Lack of the co-authors (Xue and Yang) in reference (“Wang, 2020).

       This part is due to the previous translation problems,We have used numbers to mark this part. This part can be located to the 96 line of the new revised article.

2.8. (Line 117) “(Wang, Wei, 2022)” is not in References section.

       This part is due to the previous translation problems,We have used numbers to mark this part. This part can be located to the 122 line of the new revised article.

2.9. (Line 119) “Benitez, 2018” is an incorrect reference. And “Fang, 2020” cannot be found.

       This part is due to the previous translation problems,We have used numbers to mark this part. This part can be located to the 124 line of the new revised article.

2.10. (Line 128) There is not in References section (“Jinhuan, 2022”).

       This part is due to the previous translation problems,We have used numbers to mark this part. This part can be located to the 137 line of the new revised article.

2.11. (Line 140) I didn't find the text (“(Yuan Runsong, 2016”) in the journal.

       This part is due to the previous translation problems,We have used numbers to mark this part. This part can be located to the 201 line of the new revised article.

2.12. (Line 162) Wrong numbering of section. It should be “2.2”, not “1.2”. And the following sections have incorrect numbering.

       According to the expert's opinion, we have revised the chapter number of the article.

2.13. (Lines 198-200) Aghion (1992) is an important reference, but it needs updated reference(s). Furthermore, that reference is incorrect – it has two authors, Philippe Aghion and Peter Howitt.

       We have used numbers to mark this part. This part can be located to the 261 line of the new revised article.And the literature was updated according to the content.

2.14. (Lines 253-254) The equation (1) has two references, but I cannot find these texts in respectively journals. Please, review.

       This part is due to the previous translation problems,We have used numbers to mark this part. This part can be located to the 317 line of the new revised article.

2.15. (Lines 450-451) As such above, equation (2) has a reference (“Wang, Ruchi, 2021”) I cannot find.

       This part is due to the previous translation problems,We have used numbers to mark this part. This part can be located to the 511 line of the new revised article.

       Perhaps due to the limitations of the model of the paper itself, it may not adequately meet the requirements of the teacher. We hope that the relevant revisions can be recognized by experts.and also hope you find our manuscript of interest and look forward to hearing from you soon. Thank you very much.

Sincerely,

Jie Li, Shengjun Yuan and Jun Wu

Reviewer 2 Report

The theme A study on the Promotional Effect and Mechanism of National 2E-Commerce Demonstration City Construction on Green Innovation Capacity of Cities  is worthy of investigation. However, the following needs to be addressed.

1. Abstract should clearly state the essence of the problem you are addressing, what you did and what you found and recommend.

2. Please modify the format and method of literature reference according to the requirements of the journal. Duan, Shixia, 2022 should be modified to Duan, 2022.The Introduction should have an explanation of the gaps both in research and practice.

3. Why is it timeliness to explore such a study? What it requires, what are the digital green innovation and e-commerce demonstration, some recent issue highlights the importance. See the following: Enhancing Digital Innovation for the Sustainable Transformation of Manufacturing Industry: A Pressure-State-Response System Framework to Perceptions of Digital Green Innovation and Its Performance for Green and Intelligent Manufacturing. https://doi.org/10.3390/systems10030072

4. The introduction should clarify the novelty and contribution of this study.

5. 162 Line 1.2 should be modified to 2.2. The following serial number should also be modified. Please check all serial numbers.

6. Please add a research framework in Part 2.

7. Please add a discussion section to show that the results of this study differ from the existing literature.

8. Please merge Part 6 and Part 7.

Author Response

Dear Editor,

       We are glad that the article can be reviewed by the experts and received valuable opinions from the experts. We carefully studied the suggestions of the teachers and tried our best to improve the overall writing of the thesis. Next, based on the suggestions of the experts, we will state how we have revised and the current revision situation.

  1. The overall situation of the article revision:

       We comprehensively considered the opinions of various experts. On the one hand, we reconsidered the structure of the article, and on the other hand, we revised and supplemented the missing details.

  1. Details of article revisions:

       We have made some adjustments to the structure of the article and the content. All references in the article are numbered.

2.1. Abstract should clearly state the essence of the problem you are addressing, what you did and what you found and recommend.

       We reviewed the summary and adjusted and revised it.

The problem to be addressing are: In order to explore the impact of the construction of national e-commerce demonstration cities on urban green innovation, this paper aimed to theoretically analyze the potential relationship between the two and the transmission mechanism.

what we do :we have examined the panel data formed by 297 prefecture level cities in China from 2005 to 2018 to explore the implementation effect of the pilot policies by using difference-in-differences, and carried out a series of robustness tests

what we found: (i) The pilot policy of e-commerce demonstration cities exhibited a significant promotion effect on the green innovation capacity of the different cities, and in general, the promotion effect of the pilot policy is dynamically sustainable. (ii) Analysis of the further influence mechanism showed that the pilot policy could effectively promote the development of urban green innovation capacity by enhancing the level of urban informatization, thereby attracting the concentration of scientific and technological talents. This in turn can facilitate urban innovation and entrepreneurship environment, among which the boosting effect of optimizing the urban innovation and entrepreneurship environment was the greatest. (iii) In terms of heterogeneity, the pilot policies showed significant positive effects on all the regions to which the cities belong, while the boosting effects were more significant for the cities with large populations, non-central cities, and the general science and education cities.

what we recommend:The findings of this study not only enriches the research results in the field of urban green innovation, but also has clear policy implications, which can provide useful guidance and reference value for the work of relevant departments.

2.2. Please modify the format and method of literature reference according to the requirements of the journal. Duan, Shixia, 2022 should be modified to Duan, 2022.The Introduction should have an explanation of the gaps both in research and practice.

       we have modified the format and method of literature reference according to the requirements of the journal.

       the gaps both in research and practice:This part can be located to the two sections (118-161).

it can be summarized as follows:

(1). There is a lack of research on the two, and there is a lack of looking at green innovation development from the urban level.

(2). Existing studies still can not provide quantitative analysis and empirical evidence of the research between them.

(3). There is still no unified conclusion on the relationship between e-commerce and green innovation in the existing research.

2.3. Why is it timeliness to explore such a study? What it requires, what are the digital green innovation and e-commerce demonstration, some recent issue highlights the importance.

       Why is it timeliness to explore such a study?

       The main reasons are as follows: first, due to the current situation of environment and resources; second, the state has always attached importance to the issue of green innovation; third, the issue of green innovation has also received the attention of scholars, but it has not paid attention to the impact of e-commerce related policies on Urban green innovation, and there is a lack of quantitative research. We look forward to seeing the policies related to e-commerce in a timely manner from this research, and how effective it is, so as to provide a theoretical basis for the implementation of national policies

       What it requires, what are the digital green innovation and e-commerce demonstration, some recent issue highlights the importance?

       As for the explanation of green innovation, we give a specific explanation in the fourth paragraph of the introduction. As for the explanation of e-commerce, it is more a kind of urban policy and a policy of the state to promote the integration of e-commerce and various industries in the city.The importance of problem research is mainly based on the background of the country's pursuit of low-carbon, green and innovation on the one hand, and on the other hand, it helps to clarify and explain the uncertainties existing in existing research and scientifically evaluate the policy effect, so as to better implement this policy and save various resources.

2.4. The introduction should clarify the novelty and contribution of this study.

       This part can be located in the sixth paragraph (143-161) of the introduction, and also explained in the discussion part.. the innovation of the article can be mainly classified as the following points:

(1) Looking at Green Innovation from the urban level, it is the first time to combine the research on the policy effect of e-commerce with urban green innovation

(2) Quantify the relationship between the two and conduct heterogeneity analysis to make the research results more detailed

(3) In the transmission mechanism analysis, it is the first time to creatively analyze from the path of urban informatization

2.5. 162 Line 1.2 should be modified to 2.2. The following serial number should also be modified. Please check all serial numbers.

       We have rearranged the contents of the articles.

2.6. Please add a research framework in Part 2.

       We adjusted the content of the article and added chapters 2.1 and 2.2. The original part is placed in the introduction.We adjusted the content of the article and added chapters 2.1 and 2.2. The original part is placed in the introduction. The reason for the adjustment is that the original part can better convey information and be more logical in the introduction.We have added a figure 1 to the article, which shows the content structure and analysis framework of the article.

2.7. Please add a discussion section to show that the results of this study differ from the existing literature.

       We have add the seventh part to discuss the similarities and differences between the research in this paper and the existing research.The differences can be more specifically reflected in the sixth part of the introduction.For example, compared with the existing research, we study the transmission path of e-commerce policies from the perspective of urban informatization for the first time.

2.8. Please merge Part 6 and Part 7.

       According to the opinions of experts, we merged the contents and marked them.

       Perhaps due to the limitations of the model of the paper itself, it may not adequately meet the requirements of the teacher. We hope that the relevant revisions can be recognized by experts.and also hope you find our manuscript of interest and look forward to hearing from you soon. Thank you very much.

Sincerely,

Jie Li, Shengjun Yuan and Jun Wu

Round 2

Reviewer 1 Report

The article is a good research work (topics, methodology, results and so on).

However, I detected a very important problem that needs to be overcome to publish it. There are many quoted references in the article that I cannot find in respective journals. I did not find some journals too. For example, the following references 1, 3, 5, 6, 9, 10, 12, 17, 27 and 31. I highlight the references 10, 12 and 27, related to equations 1 and 2, I would like to verify the former equations.

The coherence between bibliography and the content of paper is fundamental to be reviewed.

Author Response

Dear Editor,

       We are glad that the article can be reviewed by the experts and received valuable opinions from the experts. We carefully studied the suggestions of the teachers and tried our best to improve the overall writing of the thesis. Next, based on the suggestions of the experts, we will state how we have revised and the current revision situation.

       First of all, We would like to thank the experts for your overall recognition of our article. As for the problem of missing references.We believe that there is no problem with the source of the literature. We believe that it may be due to the differences between domestic and foreign databases. Among them, the references that experts fail to access are mainly from several academic database platforms, such as China National Knowledge Infrastructure platform,China Wanfang Data knowledge service platform,Baidu academic journal,VIP Information Chinese journal service platform,and National Social Sciences Database. For convenient search and reference, we attached the official website of the above database platform:

       China National Knowledge Infrastructure:

              https://www.cnki.net/

       China Wanfang Data knowledge service platform:

              https://www.wanfangdata.com.cn/index.html?index=true

       Baidu academic journal:

              https://xueshu.baidu.com/

       VIP Information Chinese periodical service platform:

              http://vip.hbdlib.cn/

       National Social Sciences Database:

              http://www.nssd.cn/

       Secondly, for the references in the article, we will attach DOI links to each article as much as possible (the complete DOI format is: https://doi.org/xxxx , abbreviated below for simplicity); If the DOI link is not found, we will provide the source address of the article. So as to ensure the academic, scientific, authentic and effective source of the reference source. The details are as follows:

  1. Xu Xianchun, Ren Xue, Chang Zihao Big data and green development [J] China industrial economy, 2019, (4):5-22.

DOI:10.19581/j.cnki.ciejournal.2019.04.001.

  1. Li Jinyu, Li Zeyu, Li Chao. Empirical Study on Urban Green Innovation Efficiency -- Evidence from urban agglomeration in the middle reaches of the Yangtze River [J]. Journal of Jiangxi University of Finance and economics, 2016 (06):3-16.

DOI:10.13676/j.cnki.cn36-1224/f.2016.06.001.

  1.  

  1.  
  2.  
  3.  
  4. Gao Changchun,Zou Yao. Has e-commerce promoted industrial structure transformation? --Evidence from national e-commerce demonstration cities[J]. Business Economics,2021,40(10):132-142.

DOI:10.13529/j.cnki.enterprise.economy.2021.10.014.

  1.  
  2.  
  3. Huang J W, Li Y H. How resource alignment moderates the relationship between environmental innovation strategy and green innovation performance[J ]. Journal of Business & Industrial Marketing, 2018.

DOI:10.1108/JBIM-10-2016-0253.

  1. Li Zheng, Liu Fengshuo. Can pilot innovative cities enhance urban green innovation[J]. Social Science Research,2021(04):91-99.

DOI:10.3969/j.issn.1000-4769.2021.04.010.

  1. Luo Chao-Ping,Zhu Pei-Wei,Zhang Can-Can,Chen Wen. Does the agglomeration of productive services promote urban green innovation based on the "local-neighborhood" effect[J]. Journal of Southwest University (Social Science Edition),2022,48(01):97-112.

DOI:10.13718/j.cnki.xdsk.2022.01.009.

  1. Wang J, Xue Y, Yang J. Boundary-spanning search and firms' green innovation: The moderating role of resource orchestration capability[J]. Business Strategy and the Environment, 2020, 29(2): 361-374.

DOI:10.1002/bse.2369

  1. Duan SH, Jin YL. Spatial spillover effects of green technology innovation on carbon productivity: The moderating role of local government competition[J]. Industrial Technology Economics,2022,41(02):62-69.

DOI:10.3969/j.issn.1004-910X.2022.02.007.

  1.  
  2. Xiong Yunbiao, Zhang Zixuan. The impact of green innovation on high-quality economic development under the threshold of human capital and its regional differences[J]. Ecological Economics,2022,38(04):43-52.

Source:https://d.wanfangdata.com.cn/periodical/ChlQZXJpb2RpY2FsQ0hJTmV3UzIwMjIwNzE5Eg1zdGpqMjAyMjA0MDA5Ggg2YXJnYjg0dA%3D%3D.

  1. Deng YP, Wang L, Zhou WJ. Does environmental regulation promote green innovation capacity? --Empirical evidence from China[J]. Statistical Research,2021,38(07):76-86.

DOI:10.19343/j.cnki.11-1302/c.2021.07.006.

  1. Wang Xiaoqi,Hao Shuangguang,Zhang Junmin. The New Environmental Protection Law and Green Innovation of Enterprises: "Pushing Back" or "Squeezing Out"? [J]. China Population-Resources and Environment, 2020,30(07):107-117.

DOI:10.12062/cpre.20200130.

  1. Xu Jia,Cui Jingbo. Low carbon cities and corporate green technology innovation[J]. China Industrial Economy,2020(12):178-196.

DOI:10.19581/j.cnki.ciejournal.2020.12.008.

  • Feng Y , Wang X, Du W, et al. Effects of Environmental Regulation and FDI on Urban Innovation in China: A
  1. Spatial Durbin Econometric Analysis[J]. Journal of Cleaner Production, 2019, 235: 210-224.

DOI:10.1016/j.jclepro.2019.06.184

  1. Zhang, F., Shao, J., Zhou, L.. Urban green innovation effects of environmental decentralization[J]. China Population-Resources and Environment,2021,31(12):83-92.

DOI:10.12062/cpre.20210808.

  1. Wang W, Xie Xionghui. Research on the impact of Internet use on public perceptions of environmental governance effectiveness[J]. Statistics and Information Forum,2022,37(03):108-117.

DOI:10.3969/j.issn.1007-3116.2022.03.011.

  1. Benitez J , Yang C , Teo T , et al. Evolution of the impact of e-business technology on operational competence and firm profitability: A panel data investigation[J]. Information & Management, 2018, 55(1):120-130.

DOI:10.1016/j.im.2017.08.002

  1. Fang M,Wu Y. A study on the association between relationship resources, e-commerce capabilities and firm performance[J]. Review of Economics and Management,2020,36(02):56-66.

DOI:10.13962/j.cnki.37-1486/f.2020.02.006.

  1. Yi L,Thomas H R.E-business and Sustainable Development[J].International Journal of Environment and Sustainable Development,2006,5(3):262-274.

DOI:10.1504/IJESD.2006.010896.

  1. Sui D Z,Rejeski D W.Environmental Impacts of the Emerging Digital Economy: The E-for-Environment E-Commerce?[J].Environmental Management, 2002, 29(2):155-163.

DOI:10.1007/s00267-001-0027-X

  1. Zhou Kexuan, Yu Linhui. Research on the impact of urban e-commerce transformation on FDI and its mechanism: empirical evidence from the pilot national e-commerce demonstration cities[J]. Learning and Practice,2021(12):82-92.

DOI:10.19624/j.cnki.cn42-1005/c.2021.12.015.

  1. Jin Huan,Yu Lihong,Wei Jia Li. Research on the impact of national e-commerce demonstration city construction on enterprises' green technology innovation and mechanism [J/OL]. Science and Technology Progress and Countermeasures,2022,39(10):81-90.

DOI:10.6049/kjjbydc.C202106128.

  1. Liu Naiquan, Deng Min, Cao Xigang. Does the e-commerce transformation of cities promote green and high-quality development? -- A quasi-natural experiment based on the construction of national e-commerce demonstration cities[J]. Finance and Economics Research,2021,47(04):49-63.

DOI:10.16538/j.cnki.jfe.20201115.401.

  1. Yu L. P.,Zhong C. B.. Research on the impact of innovation policies and R&D subsidies on corporate R&D investment - mechanism, magnitude and law[J]. Mathematical Statistics and Management,2020,39(06):1060-1072.

DOI:10.13860/j.cnki.sltj.20200123-007.

  1. Yuan Runsong, Feng Chao, Wang Miao, Huang Jianbai. Technological innovation, technology gap and regional green development in China[J]. Scientology Research,2016,34(10):1593-1600.

DOI:10.16192/j.cnki.1003-2053.2016.10.019.

  1. Wang P,Xie LW. Pollution control investment, enterprise technology innovation and pollution control efficiency[J]. China Population-Resources and Environment,2014,24(09):51-58.

DOI:10.3969/j.issn.1002-2104.2014.09.008.

  1. Shao Shuai,Zhang Ke,Dou Jianmin. Energy saving and emission reduction effects of economic agglomeration:theory and Chinese experience[J]. Management World,2019,35(01):36-60.

DOI:10.3969/j.issn.1002-5502.2019.01.004.

  1. Wang Ruqi,Hu Xuhua. The impact of the integration strategy of Yangtze River Economic Belt on the innovation capacity of cities[J]. East China Economic Management,2021,35(10):29-38.

DOI:10.19629/j.cnki.34-1014/f.210604001.

  1. Shi Dachian, Ding Hai, Wei Ping, Liu Jianjiang. Can smart city construction reduce environmental pollution[J]. China industrial economy,2018(06):117-135.

DOI:10.19581/j.cnki.ciejournal.2018.06.008.

  1. Huo L,Ning N. Research on the dynamic mechanism of the impact of Internet development on regional innovation efficiency[J]. Journal of Northwestern University (Philosophy and Social Science Edition),2020,50(03):144-156.

DOI:10.16152/j.cnki.xdxbsk.2020-03-016.

  1. Han Xianfeng,Hui Ning,Song Wenfei. Can informatization improve the efficiency of technological innovation in China's industrial sector[J]. China Industrial Economics,2014(12):70-82.

DOI:10.19581/j.cnki.ciejournal.2014.12.006.

  1. Jiang Yanpeng, Wang Xinjing, Ma Renfeng. Theoretical exploration of innovative talent concentration -- the perspective of city selection of global talent flow [J]. Geographic science, 2021,41 (10): 1802-1811.

DOI:10.13249/j.cnki.sgs.2021.10.012.

  1. Yuan H,Zhu Chengliang. Have national high-tech zones promoted the transformation and upgrading of China's industrial structure[J]. China industrial economy,2018(08):60-77.

DOI:10.19581/j.cnki.ciejournal.2018.08.004.

  1. Wang M,Xuan Y,Chen Qifei. Creative class agglomeration, knowledge externalities and urban innovation--Evidence from 20 large cities[J]. Economic Theory and Economic Management,2016(01):59-70.

DOI:10.3969/j.issn.1000-596X.2016.01.003.

  1.  
  2. Li Yuting, Zhu Zhiyong. Institutional supply and green innovation efficiency in Chinese regions[J]. Journal of Beijing University of Technology (Social Science Edition),2019,21(01):50-58.

DOI:10.15918/j.jbitss1009-3370.2019.5347.

  1. Chen Huaichao,Zhang Jing,Fei Yuting. Does institutional support promote collaborative innovation between industry, university and research? -The moderating role of firms' absorptive capacity and the mediating role of industry-university-research cooperation tightness[J]. Science Research Management,2020,41(03):1-11.

DOI:10.19571/j.cnki.1000-2995.2020.03.001.

  1. Peng Heng, Li Yang. Intellectual property protection and green total factor productivity in China[J]. Economic system reform,2019(03):18-24.

Source:https://d.wanfangdata.com.cn/periodical/ChlQZXJpb2RpY2FsQ0hJTmV3UzIwMjIwNzE5Eg9qanR6Z2cyMDE5MDMwMDQaCHBqOGZ2dmxu

  1. Bai Junhong,Jiang Fuxin. Collaborative innovation, spatial association and regional innovation performance[J]. Economic Research,2015,50(07):174-187.

Source:https://d.wanfangdata.com.cn/periodical/ChlQZXJpb2RpY2FsQ0hJTmV3UzIwMjIwNzE5Eg1qanlqMjAxNTA3MDEzGgg1aHRhNjRweA%3D%3D

       Finally, we rechecked the citations of the literature and we corrected one citation error.For the citation of equation 1, we checked the citation of the article and found no errors. For the citation of equation 2, we found a citation error and we modified it. The cited literature 27 was modified to 32.

       Perhaps due to the limitations of the model of the paper itself, it may not adequately meet the requirements of the teacher. We hope that the relevant revisions can be recognized by experts.and also hope you find our manuscript of interest and look forward to hearing from you soon. Thank you very much.

Sincerely,

Jie Li, Shengjun Yuan and Jun Wu

Reviewer 2 Report

The authors revised the manuscript carefully. However, authors need to modify the reference format according to the requirements of the journal.

Author Response

Dear Editor,

       We are glad that the article can be reviewed by the experts and received valuable opinions from the experts. We carefully studied the suggestions of the teachers and tried our best to improve the overall writing of the thesis. Next, based on the suggestions of the experts, we will state how we have revised and the current revision situation.

       We would like to thank the experts for your overall recognition of our article,For the References section,We added DOI links to the References section according to the format of the journal,we will attach DOI links to each article as much as possible (the complete DOI format is: https://doi.org/xxxx , abbreviated below for simplicity); If the DOI link is not found, we will provide the source address of the article. So as to ensure the academic, scientific, authentic and effective source of the reference source. The details are as follows:

  1. Xu Xianchun, Ren Xue, Chang Zihao Big data and green development [J] China industrial economy, 2019, (4):5-22.

DOI:10.19581/j.cnki.ciejournal.2019.04.001.

  1. Li Jinyu, Li Zeyu, Li Chao. Empirical Study on Urban Green Innovation Efficiency -- Evidence from urban agglomeration in the middle reaches of the Yangtze River [J]. Journal of Jiangxi University of Finance and economics, 2016 (06):3-16.

DOI:10.13676/j.cnki.cn36-1224/f.2016.06.001.

  1.  

  1.  
  2.  
  3.  
  4. Gao Changchun,Zou Yao. Has e-commerce promoted industrial structure transformation? --Evidence from national e-commerce demonstration cities[J]. Business Economics,2021,40(10):132-142.

DOI:10.13529/j.cnki.enterprise.economy.2021.10.014.

  1.  
  2.  
  3. Huang J W, Li Y H. How resource alignment moderates the relationship between environmental innovation strategy and green innovation performance[J ]. Journal of Business & Industrial Marketing, 2018.

DOI:10.1108/JBIM-10-2016-0253.

  1. Li Zheng, Liu Fengshuo. Can pilot innovative cities enhance urban green innovation[J]. Social Science Research,2021(04):91-99.

DOI:10.3969/j.issn.1000-4769.2021.04.010.

  1. Luo Chao-Ping,Zhu Pei-Wei,Zhang Can-Can,Chen Wen. Does the agglomeration of productive services promote urban green innovation based on the "local-neighborhood" effect[J]. Journal of Southwest University (Social Science Edition),2022,48(01):97-112.

DOI:10.13718/j.cnki.xdsk.2022.01.009.

  1. Wang J, Xue Y, Yang J. Boundary-spanning search and firms' green innovation: The moderating role of resource orchestration capability[J]. Business Strategy and the Environment, 2020, 29(2): 361-374.

DOI:10.1002/bse.2369

  1. Duan SH, Jin YL. Spatial spillover effects of green technology innovation on carbon productivity: The moderating role of local government competition[J]. Industrial Technology Economics,2022,41(02):62-69.

DOI:10.3969/j.issn.1004-910X.2022.02.007.

  1.  
  2. Xiong Yunbiao, Zhang Zixuan. The impact of green innovation on high-quality economic development under the threshold of human capital and its regional differences[J]. Ecological Economics,2022,38(04):43-52.

Source:https://d.wanfangdata.com.cn/periodical/ChlQZXJpb2RpY2FsQ0hJTmV3UzIwMjIwNzE5Eg1zdGpqMjAyMjA0MDA5Ggg2YXJnYjg0dA%3D%3D.

  1. Deng YP, Wang L, Zhou WJ. Does environmental regulation promote green innovation capacity? --Empirical evidence from China[J]. Statistical Research,2021,38(07):76-86.

DOI:10.19343/j.cnki.11-1302/c.2021.07.006.

  1. Wang Xiaoqi,Hao Shuangguang,Zhang Junmin. The New Environmental Protection Law and Green Innovation of Enterprises: "Pushing Back" or "Squeezing Out"? [J]. China Population-Resources and Environment, 2020,30(07):107-117.

DOI:10.12062/cpre.20200130.

  1. Xu Jia,Cui Jingbo. Low carbon cities and corporate green technology innovation[J]. China Industrial Economy,2020(12):178-196.

DOI:10.19581/j.cnki.ciejournal.2020.12.008.

  • Feng Y , Wang X, Du W, et al. Effects of Environmental Regulation and FDI on Urban Innovation in China: A
  1. Spatial Durbin Econometric Analysis[J]. Journal of Cleaner Production, 2019, 235: 210-224.

DOI:10.1016/j.jclepro.2019.06.184

  1. Zhang, F., Shao, J., Zhou, L.. Urban green innovation effects of environmental decentralization[J]. China Population-Resources and Environment,2021,31(12):83-92.

DOI:10.12062/cpre.20210808.

  1. Wang W, Xie Xionghui. Research on the impact of Internet use on public perceptions of environmental governance effectiveness[J]. Statistics and Information Forum,2022,37(03):108-117.

DOI:10.3969/j.issn.1007-3116.2022.03.011.

  1. Benitez J , Yang C , Teo T , et al. Evolution of the impact of e-business technology on operational competence and firm profitability: A panel data investigation[J]. Information & Management, 2018, 55(1):120-130.

DOI:10.1016/j.im.2017.08.002

  1. Fang M,Wu Y. A study on the association between relationship resources, e-commerce capabilities and firm performance[J]. Review of Economics and Management,2020,36(02):56-66.

DOI:10.13962/j.cnki.37-1486/f.2020.02.006.

  1. Yi L,Thomas H R.E-business and Sustainable Development[J].International Journal of Environment and Sustainable Development,2006,5(3):262-274.

DOI:10.1504/IJESD.2006.010896.

  1. Sui D Z,Rejeski D W.Environmental Impacts of the Emerging Digital Economy: The E-for-Environment E-Commerce?[J].Environmental Management, 2002, 29(2):155-163.

DOI:10.1007/s00267-001-0027-X

  1. Zhou Kexuan, Yu Linhui. Research on the impact of urban e-commerce transformation on FDI and its mechanism: empirical evidence from the pilot national e-commerce demonstration cities[J]. Learning and Practice,2021(12):82-92.

DOI:10.19624/j.cnki.cn42-1005/c.2021.12.015.

  1. Jin Huan,Yu Lihong,Wei Jia Li. Research on the impact of national e-commerce demonstration city construction on enterprises' green technology innovation and mechanism [J/OL]. Science and Technology Progress and Countermeasures,2022,39(10):81-90.

DOI:10.6049/kjjbydc.C202106128.

  1. Liu Naiquan, Deng Min, Cao Xigang. Does the e-commerce transformation of cities promote green and high-quality development? -- A quasi-natural experiment based on the construction of national e-commerce demonstration cities[J]. Finance and Economics Research,2021,47(04):49-63.

DOI:10.16538/j.cnki.jfe.20201115.401.

  1. Yu L. P.,Zhong C. B.. Research on the impact of innovation policies and R&D subsidies on corporate R&D investment - mechanism, magnitude and law[J]. Mathematical Statistics and Management,2020,39(06):1060-1072.

DOI:10.13860/j.cnki.sltj.20200123-007.

  1. Yuan Runsong, Feng Chao, Wang Miao, Huang Jianbai. Technological innovation, technology gap and regional green development in China[J]. Scientology Research,2016,34(10):1593-1600.

DOI:10.16192/j.cnki.1003-2053.2016.10.019.

  1. Wang P,Xie LW. Pollution control investment, enterprise technology innovation and pollution control efficiency[J]. China Population-Resources and Environment,2014,24(09):51-58.

DOI:10.3969/j.issn.1002-2104.2014.09.008.

  1. Shao Shuai,Zhang Ke,Dou Jianmin. Energy saving and emission reduction effects of economic agglomeration:theory and Chinese experience[J]. Management World,2019,35(01):36-60.

DOI:10.3969/j.issn.1002-5502.2019.01.004.

  1. Wang Ruqi,Hu Xuhua. The impact of the integration strategy of Yangtze River Economic Belt on the innovation capacity of cities[J]. East China Economic Management,2021,35(10):29-38.

DOI:10.19629/j.cnki.34-1014/f.210604001.

  1. Shi Dachian, Ding Hai, Wei Ping, Liu Jianjiang. Can smart city construction reduce environmental pollution[J]. China industrial economy,2018(06):117-135.

DOI:10.19581/j.cnki.ciejournal.2018.06.008.

  1. Huo L,Ning N. Research on the dynamic mechanism of the impact of Internet development on regional innovation efficiency[J]. Journal of Northwestern University (Philosophy and Social Science Edition),2020,50(03):144-156.

DOI:10.16152/j.cnki.xdxbsk.2020-03-016.

  1. Han Xianfeng,Hui Ning,Song Wenfei. Can informatization improve the efficiency of technological innovation in China's industrial sector[J]. China Industrial Economics,2014(12):70-82.

DOI:10.19581/j.cnki.ciejournal.2014.12.006.

  1. Jiang Yanpeng, Wang Xinjing, Ma Renfeng. Theoretical exploration of innovative talent concentration -- the perspective of city selection of global talent flow [J]. Geographic science, 2021,41 (10): 1802-1811.

DOI:10.13249/j.cnki.sgs.2021.10.012.

  1. Yuan H,Zhu Chengliang. Have national high-tech zones promoted the transformation and upgrading of China's industrial structure[J]. China industrial economy,2018(08):60-77.

DOI:10.19581/j.cnki.ciejournal.2018.08.004.

  1. Wang M,Xuan Y,Chen Qifei. Creative class agglomeration, knowledge externalities and urban innovation--Evidence from 20 large cities[J]. Economic Theory and Economic Management,2016(01):59-70.

DOI:10.3969/j.issn.1000-596X.2016.01.003.

  1.  
  2. Li Yuting, Zhu Zhiyong. Institutional supply and green innovation efficiency in Chinese regions[J]. Journal of Beijing University of Technology (Social Science Edition),2019,21(01):50-58.

DOI:10.15918/j.jbitss1009-3370.2019.5347.

  1. Chen Huaichao,Zhang Jing,Fei Yuting. Does institutional support promote collaborative innovation between industry, university and research? -The moderating role of firms' absorptive capacity and the mediating role of industry-university-research cooperation tightness[J]. Science Research Management,2020,41(03):1-11.

DOI:10.19571/j.cnki.1000-2995.2020.03.001.

  1. Peng Heng, Li Yang. Intellectual property protection and green total factor productivity in China[J]. Economic system reform,2019(03):18-24.

Source:https://d.wanfangdata.com.cn/periodical/ChlQZXJpb2RpY2FsQ0hJTmV3UzIwMjIwNzE5Eg9qanR6Z2cyMDE5MDMwMDQaCHBqOGZ2dmxu

  1. Bai Junhong,Jiang Fuxin. Collaborative innovation, spatial association and regional innovation performance[J]. Economic Research,2015,50(07):174-187.

Source:https://d.wanfangdata.com.cn/periodical/ChlQZXJpb2RpY2FsQ0hJTmV3UzIwMjIwNzE5Eg1qanlqMjAxNTA3MDEzGgg1aHRhNjRweA%3D%3D

       Perhaps due to the limitations of the model of the paper itself, it may not adequately meet the requirements of the teacher. We hope that the relevant revisions can be recognized by experts.and also hope you find our manuscript of interest and look forward to hearing from you soon. Thank you very much.

Sincerely,

Jie Li, Shengjun Yuan and Jun Wu

Round 3

Reviewer 1 Report

Thank you very much for answers. Congratulations.